# Chondroblastic Osteosarcoma Associated with Previous Chronic Osteomyelitis Caused by *Serratia liquefaciens* in a German Shepherd Dog

**DOI:** 10.3390/vetsci9030096

**Published:** 2022-02-22

**Authors:** Delia Franchini, Serena Paci, Stefano Ciccarelli, Carmela Valastro, Grazia Greco, Antonio Di Bello

**Affiliations:** Department of Veterinary Medicine, University of Bari “Aldo Moro”, SP 62 Km 3, 70010 Valenzano, Italy; delia.franchini@uniba.it (D.F.); serena.paci@uniba.it (S.P.); carmela.valastro@uniba.it (C.V.); grazia.greco@uniba.it (G.G.); antonio.dibello@uniba.it (A.D.B.)

**Keywords:** bone tumor, chondroblastic osteosarcoma, chronic osteomyelitis, *Serratia liquefaciens*

## Abstract

Development of bone tumors as a result of chronic osteomyelitis represents a relatively rare and late complication in humans and animals. We described a malignant transformation (chondroblastic osteosarcoma) in a 7-year-old German shepherd with a history of polyostotic osteomyelitis caused by *Serratia liquefaciens* when the dog was 15 months old. The tumor developed in the right humeral diaphysis, one of the sites of polyostotic osteomyelitis. To the best of our knowledge this is the first report of polyostotic osteomyelitis caused by *Serratia liquefaciens* in dogs.

## 1. Introduction

Septic polyostotic osteomyelitis is a rare bone infection in dogs, involving multiple bones and commonly caused by a long-term or persistent infection with fungal or bacterial agents [1,2,3,4,5,6,7]. Generally, a wide range of microorganisms could cause osteomyelitis, with bacterial infections being most frequently reported [8]. However, polyostotic osteomyelitis is more likely to be fungal than bacterial [8]. Many studies suggest *Staphylococcus* spp. to be the most common microorganism to cause osteomyelitis in companion animals, although other Gram-positive bacteria, including *Streptococcus*, and Gram-negative bacteria (e.g., *E. coli*, *Pseudomonas* sp., *Proteus* sp., *Pasteurella multocida* and *Klebsiella* sp.) [9,10] have been associated with the disease. Recently, anaerobic bacteria have also been identified (e.g., *Propionibacterium acnes*, *Gemella morbillorum*, *Bacteroides* and *Fusobacterium* sp.) [6]. In a study carried out to detect the aerobic bacteria, which are responsible for post-traumatic infection before and after fracture, authors reported osteomyelitis caused by *Serratia liquefaciens* [11], an opportunistic Gram-negative, facultatively anaerobic of the family Yersiniaceae. *Serratia* spp. infections (especially *S. marcescens*, *S. liquefaciens* and *S. plymuthica*) have also been reported in certain groups of people, such as intravenous drug users, immunocompromised patients, and those who have undergone invasive procedures [12]. Bone polyostotic lesions are likely to be aggressive and have a narrow list of differentials: primary or secondary neoplasia, septic or aseptic osteomyelitis or immune-mediated diseases. Differentiating malignant bone neoplasia from osteomyelitis can be difficult, even if the radiographic pattern is different, and more rapid bone changes tend to occur with neoplasia than osteomyelitis [13]. Chronic and persistent inflammation promotes cancer development and predisposes to carcinogenesis [14,15,16,17,18]. Malignant transformation as a result of chronic osteomyelitis represents a relatively rare and late complication. The mechanism of malignant transformation remains unknown. It is assumed that the chronic inflammatory state behaves as a promoter in the complex process of carcinogenesis [19].

In this report we described a septic polyostotic osteomyelitis sustained by *Serratia liquefaciens* in a 15-month-old female German shepherd that developed chondroblastic osteosarcoma (OSA) of the right humerus six years later, previously the site of septic osteomyelitis.

## 2. Case Presentation

A 15-month-old entire female, German shepherd dog was admitted to the Teaching Hospital of the Department of Veterinary Medicine at University of Bari (THDVM) due to a weight-bearing lameness of the left forelimb and weakness. As reported by the owner, four months earlier the animal had suffered from epistaxis, depression, hyperthermia and II-degree lameness of the left forelimb, and blood tests showed a slight leukopenia, normocytic hypochromic anemia, an increase platelet volume and macroplatelets, a marked increase in phosphorus and C-reactive protein (CRP) (4.12 mg/dL, normal range 0.01–0.45 mg/dL), a mild increase in globulin α1 and α2 and monoclonal hyper-γ-globulinemia. The dog, resulting positive (1:320) for *Erlichia canis* (immunofluorescence–IFA) and negative for *Leishmania infantum* (ELISA) by serological test, was treated with doxycycline at 10 mg/Kg PO SID for 21 days, becoming serologically negative for *Erlichia* sp. Due to the worsening of the lameness, the dog was referred to the THDVM. On clinical examination, the dog had no hyperthermia (38.7 °C), and exhibited signs of pain on deep palpation together with soft tissue swelling of the left humerus and bilateral hypomyotrophy of both forelimbs. Laboratory analyses highlighted a marked hypochromic normocytic anemia, lymphocytosis (3225/μL), eosinophilia (1032/μL), persistence of high CRP and hyperphosphatemia. Radiographic examination showed multiple bone lesions that suggested polyostotic osteomyelitis involving the left and right humeral diaphysis, the left proximal epiphysis of tibia and 5th, 6th and 7th right and left ribs. Bone sclerosis and a widespread periosteal reaction, with spicules associated with cortical thickening and exostosis, were evident in medio-lateral and cranio-caudal projections of left and right humeral diaphysis (Figure 1a). The foramen of the nutritional artery appeared increased and the spongiosa showed areas of bone thickening, mixed with more radiolucent areas of rarefaction. However, radiographic changes in the right humerus were less marked than in the left (Figure 1b). The radiographic examination in medio-lateral and cranio-caudal views of the hind limbs did not show signs compatible with osteomyelitis at the right limb, while the left proximal epiphysis of tibia showed an increase of the foramen of the nutritional artery. Orthogonal views of the thorax (left and right lateral and dorsoventral) showed focal periosteal new bone affecting the right 5th, 6th, and 7th and left 6th ribs (Figure 2). The dog was treated with amoxicillin/clavulanic acid 25 mg/Kg per os (PO) BID for three weeks, carprofen at 2 mg/Kg PO BID and pantoprazole at 1 mg/Kg PO SID for 10 days, with evident clinical improvement. One month later, the dog presented severe weakness and a left humeral fistula that had a purulent discharge. Blood and urinary samples resulted negative to microbiological investigations. A surgical biopsy from the lytic area of the left and right humerus was performed and histological findings showed non-specific chronic inflammation. Bone marrow aspirates samples tested positive for *Ehrlichia* sp. using polymerase chain reaction (PCR). Fine-needle aspirates (FNAs) of the left suprascapular lymph node revealed septic pyogranulomatous inflammation with mild reactive lymphoid hyperplasia and abnormal neutrophils. Mycological and bacteriological investigations were carried out on the bone biopsy and *S. liquefaciens* sensitive to amikacin, ceftazidime, enrofloxacin and ciprofloxacin was isolated. Enrofloxacin at 5 mg/Kg PO BID for 60 days, doxycycline at 10 mg/Kg PO SID for 42 days, gabapentin at 10 mg/Kg for 30 days PO SID, meloxicam at 0.1 mg/Kg PO SID and omeprazole at 1 mg/Kg both for 10 days PO SID were administered. Sixty days later, the dog showed optimal physical condition, no lameness, and the hematological and biochemical tests were within normal values. The negativity to *Ehrlichia* sp. was confirmed by bone marrow PCR, and radiographic follow-up showed an intense bone reshaping. 

After 6 years, the dog return to the THDVM for a 2-week history of III-degree lameness of the right front limb. Physical examination showed visible hypomyotrophy on the abductor muscles and intense pain on palpation of the right forelimb while the left limb appeared normal. The medio-lateral and cranio-caudal radiographic examination of the right humeral diaphysis showed a mixture of lysis and focal sclerosis, poorly defined transition zone between the lytic lesion, marked cortex destruction, and normal bone, and irregular periosteal reaction and large presumptive neoplastic osteogenesis in the surrounding soft tissue (Figure 3a,b). Clinical and radiographic findings of the humeral lesion were compatible with bone neoplasm. The dog underwent a total body computed tomography (CT) scan to stage the suspected neoplasm. The tomographic image of the right humerus showed cortical bone destruction, amorphous periosteal productive response, with transition zones, and focal regions of hyperdensity within the medullary cavity. The pattern was compatible with a bone neoplasm of the right humerus (Figure 4a). The left humerus CT scan showed cortical bone osteolysis, regular and solid periosteal reaction as a result of a chronic osteomyelitis process. In the middle-third of the right 5th, 6th and 7th ribs and left 6th rib, a focal thickening attributable to hyperostosis was highlighted (Figure 4b). Bone biopsy of the right humeral mass and left humeral diaphysis with osteolysis, performed during CT for the histological and microbiological examination, showed chronic lymphoplasmacellular osteomyelitis with bone remodeling without identifying neoplasm or any causative agent, and hemoculture was also negative. The failure of 6 weeks of medical therapy with NSAIDs (non-steroidal anti-inflammatory drugs), gabapentin and pamidronate disodium, and the progression of the lameness led to the decision of amputating the right forelimb. By histological examination of the entire bone, a productive chondroblastic OSA of the distal right humerus with infiltration of the surrounding soft tissues was diagnosed. Histological sample showed lamellar bone tissue with irregular islands of osteoid surrounded by malignant spindle to polygonal cells, organized in carpet or short bundles supported by scarce vascular stroma with productive extracellular osteoid matrix. Cartilage differentiation was appreciated multifocally in association with large areas of necrosis. The owners refused standard adjuvant chemotherapy with carboplatin or doxorubicin. The dog underwent a metronomic chemotherapy with cyclophosphamide (15 mg/m^2^/SID) and firocoxib (5 mg/Kg/SID), and thalidomide at 10 mg/Kg, PO SID was administered. Eight months later the dog showed developed vertebral and thoracic metastases at CT scan restaging and was euthanized following the owner’s will.

## 3. Discussion

This report describes the bone colonization by *S. liquefaciens* causing polyostotic osteomyelitis in a dog that was also infected with *Ehrlichia canis*. Several studies have highlighted how *Ehrlichia*-induced immunosuppression may cause the organism to be more sensitive for secondary infections caused by bacteria, viruses, and protozoa [20], suggesting that chronic infection and uncontrolled long-term stimulation of the immune system could contribute to the pathogenesis of septic osteomyelitis. Following the diagnosis of *E. canis* infection, the initial doxycycline-based therapy was not administered for an adequate number of days, leading to a persistent systemic infection. This case is novel given the isolation of *S. liquefaciens*, because this microorganism has never been isolated in a case of polyostotic osteomyelitis. The biofilm produced by *S. liquefaciens* resulted to be protective from the humoral and cell-mediated immunity of the host, already weakened by ehrlichiosis [21]. *Serratia liquefaciens* has been isolated only in infections caused by intravenous catheters in dogs with parvovirosis, without inducing subsequent osteomyelitis [22], and in a post-traumatic infection before and after fracture osteosynthesis [11]. Human infections by members of the genus *Serratia* were not well-recognized until the latter half of the 20th century. *Serratia marcescens* is frequently isolated in human infections [23]. 

In this report the evidence of polyostotic osteomyelitis sustained by *Serratia* was also peculiar [2,3,4,5,6,7], compared to the greater evidence in the literature of monostotic osteomyelitis [24]. First-line therapy with amoxicillin/clavulanic acid was not targeted towards the specific microorganism and thus it was ineffective. Culture examination on bone biopsy was necessary to identify the etiological agent, because the antibiogram provided specific and effective antibiotic therapy. The combination of NSAIDs, analgesics and antibiotics therapy provided the remission of pain and swelling of the soft tissues. As a result of the risk factors and the history of chronic osteomyelitis in the forelimbs, the dog was highly susceptible to develop OSA [25], which is a cancer of large and giant breeds, including the German shepherd [26]. Although the etiology of OSA in dogs is unknown, even minor trauma to bone tissue can trigger the disease [27] and, in general, chronic inflammation favors carcinogenesis [28,29,30]. Physical, chemical or infectious lesions trigger a sequence of events that constitutes the inflammatory response, which if it becomes chronic can cause a possible failure to control the immune response by modifying the cellular microenvironment and thus leading to alterations in cancer-related genes [16]. Failure or non-treatment of the osteomyelitis leads to a chronic and refractory bone infection, in which constant inflammatory activity causes bone destruction and may developed the malignant transformation [16]. The interval between the original bacterial infection and malignant degeneration can be many years and often can be decades in people (ranging from 18 to 72 years). In this study, an interval of 6 years was observed between osteomyelitis diagnosis and development of osteosarcoma. Surgical amputation of the affected limb as standard therapy for the local treatment of OSA allowed a good post-operative recovery of the dog in association with adjuvant chemotherapy with a median DFI (Disease Free Interval) of 327 days and an MST (Median Survival Time) of 383 days [26]. Owners refused standard adjuvant chemotherapy after the amputation and started metronomic chemotherapy. Although chronic lymphoplasmacellular osteomyelitis was diagnosed in both humeri, the tumor has developed on one bone segment, and the development of new similar tumors cannot be excluded in other sites also affected by osteomyelitis.

## 4. Conclusions

*Serratia liquefaciens* was herein isolated for the first time from a bone sample of a dog with polyostotic osteomyelitis. Polyostotic osteomyelitis can manifest itself as an alarming condition for the animals and, if not properly treated, can lead to death. The effective treatment of this pathology can only be achieved by implementing the complete diagnostic process, together with a histological evaluation of the bone biopsies. Chronic osteomyelitis may present as recurrent disease, and periods of quiescence can be of variable duration. The incidence of relapse following an apparently successful treatment remains high. The management of chronic osteomyelitis is challenging and a multidisciplinary approach involving radiologist, microbiologist and surgeon is required. Even following long periods of antibiotic treatment and surgical debridement, exacerbations can occur for many years and the cure of the disease cannot be safely declared. In case of very aggressive osteomyelitis, especially in medium-large breed dogs, a long-term follow-up should be applied due to the risk of late tumor development.

## Figures and Tables

**Figure 1 vetsci-09-00096-f001:**
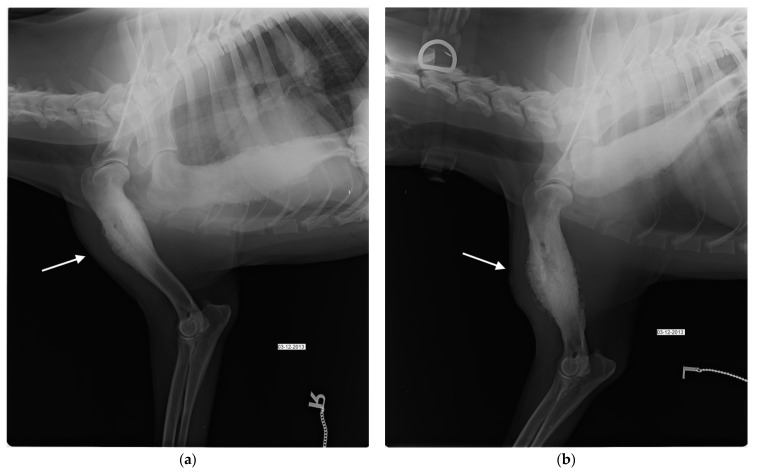
Mediolateral (ML) view of the left (**a**) and right (**b**) humerus at the day of first admission shows diffuse sclerosis and diaphyseal periosteal reaction (white arrow).

**Figure 2 vetsci-09-00096-f002:**
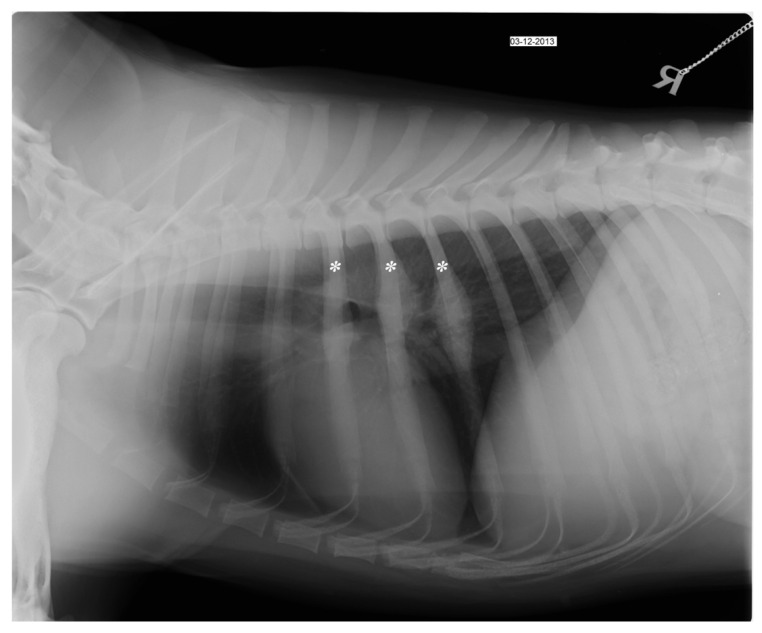
Right lateral (LL) thoracic radiograph at the day of first admission shows focal periosteal reaction of the 5th, 6th and 7th ribs (asterisk).

**Figure 3 vetsci-09-00096-f003:**
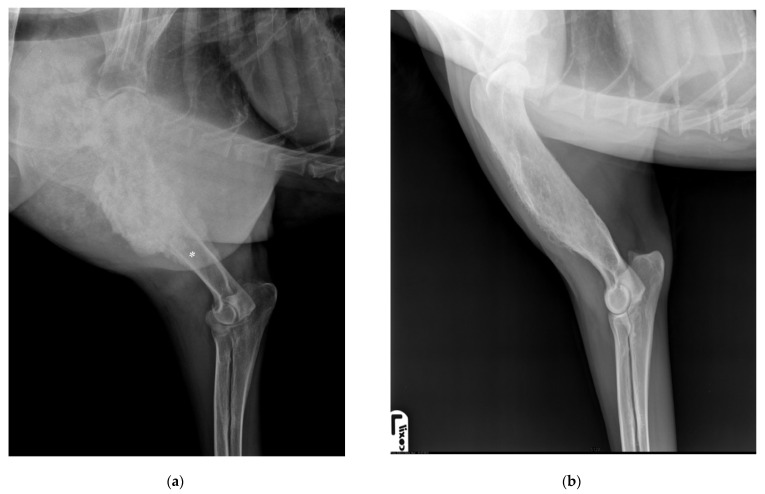
(**a**) Lateral view of right humerus, six years after the first admission, shows a heterogeneous lesion in the proximal right humeral diaphysis characterized by a mixture of lysis and focal sclerosis, and an active, irregular periosteal reaction and the indistinct transition zone proximally between normal and abnormal bone (asterisk). (**b**) Lateral view of left humerus, six years after the first admission, shows a residual periosteal reaction at the humeral diaphysis.

**Figure 4 vetsci-09-00096-f004:**
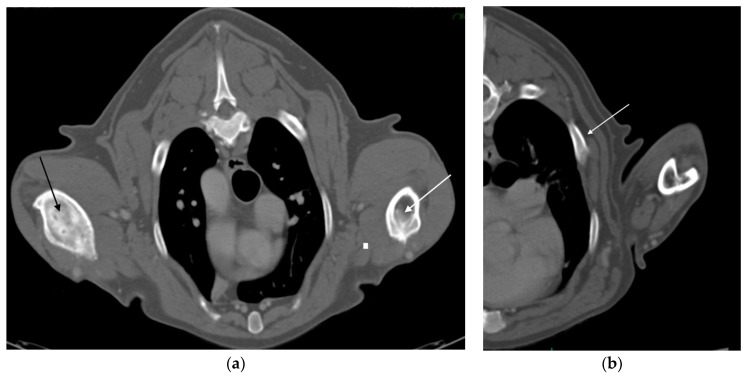
(**a**) Transverse CT image, six years after the first admission, shows a severe lysis of the right diaphyseal humerus (black arrow). On the contralateral humerus, outcomes of a previous osteomyelitis are observed, with considerable thickening of the cortex (white arrow). (**b**) Thorax transverse CT reconstruction shows focal costal thickening attributable to hyperostosis (white arrow).

## Data Availability

Not applicable.

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
