# Peer review of "Chondroblastic Osteosarcoma Associated with Previous Chronic Osteomyelitis Caused by Serratia liquefaciens in a German Shepherd Dog"

_vetsci, 2022, doi:10.3390/vetsci9030096_

Round 1

Reviewer 1 Report

good job, great relevance

There are few recods that describes a malignant transformation in German shepherd with a history of polyostotic osteomyelitis sustained by Serratia liquefaciens when the dog was 15 months old.

The case is well described and it never described a correlation with tumors.

Regarding the methodology, limb sparing is a good choice for this patient. Clinical and xray suggested tumor

The references are appropriate.

Author Response

We want to thank the reviewer for the good opinion on our paper

Reviewer 2 Report

General comment:

The discussion is a little confusing, not well structured with a mix of short sentences about different arguments. I suggest taking more care of the discussion. 

I also suggest an English review

Rows 12-13: “The tumor developed in the right humeral diaphysis one of the sites of polyostotic osteomyelitis.“ add a comma after the word diaphysis

Rows 18-19: “Septic polyostotic osteomyelitis is a rare bone infection in dog, commonly caused by a long- term or persistent infection with fungal and bacterial agents”: add the meaning of ‘polyostotic’ inside the sentence (as Septic polyostotic osteomyelitis is a rare bone infection in dog, involving multiple bones and commonly caused by a long- term or persistent infection with fungal and bacterial agents)

Rows 19-20: “a wide range of microorganisms could cause osteomyelitis, with bacterial infections being the most frequently reported”: add please a reference.

Row 23: delete “causative agent of ”

Row 26: I would write ‘with the disease’ instead of “ with osteomyelitis” to avoid the repetition.

Row 29: you should add some information about Serratia spp. as family, type of GRAM reactivity, and the natural environment for this bacteria (which represents an important nosocomial infection in humans)

Row 34: by “osteomyelitis” do you mean “infectious osteomyelitis”?

Row 48, 57, 92: the name of the hospital is missing

Row 52 and following: can you add, please, the values of the normal range when you give the results of the chemistry?

Row 56: write “Erlichia sp..”

Rows 58-59: I would write as follows “On clinical examination, the dog had no hyperthermia (38.7°C), exhibited signs of pain on deep palpation, together with soft tissue swelling, of left humerus and bilateral hypomyotrophy of both forelimbs.”

Row 64: I would write “the 5th, 6th and 7th right and left ribs” 

Row 64 (and legend of Figure 1, 3, and row 96): regarding the “sclerosis”, how can you detect and confirm that there is a deposit of fibrous tissue via X-ray? I’m not a radiologist, this is why I’m asking.

Row 64: Please, add a comma after the word ‘reaction’.

Rows 75, 76: spell out PO, BID and SID the first time you mention them in the text.

Rows 83-84: what do you mean with “abnormal neutrophils.”?

Rows 90-91: it is not clear, to me, on what biological sample the PCR was performed to prove the negativity to Ehrlichia sp.

Rows 94-95: do you mean that the left limb, not the right, appeared normal? 

Row 98: the radiograph alone can not establish that there is for sure a “large neoplastic osteogenesis”. The neoplastic nature of the lesion can be revealed and confirmed just at histology, the radiograph can eventually detect an area suggestive of neoplasia (as large presumptive neoplastic osteogenesis). Please, adjust the sentence.

Row 100: spell out CT the first time you mention it.

Row 111: spell out NSAIDs

Rows 111-113: after how much time there was “The failure of medical therapy with NSAIDs, gabapentin and pamidronate disodium, and the progression of the lameness led to the decision to amputate the right forelimb”?

Row 113: I would say “By histological examination of the entire bone”

Row 113-114: can you provide a brief description of the microscopic pattern and maybe add a picture of the histology? Eventually, the description can come only with the figure legend.

Row 116: write refused instead of refuse

Rows 115-117: reformulate, please, the sentence: “The dog undergoing a metronomic chemotherapy with cyclophosphamide (15 mg/m2/SID) and firocoxib (5 mg/Kg/SID) plus thalidomide at 10 mg/Kg, PO SID was”

Rows 117-118: I would write “8 months later the dog developed vertebral and thoracic metastases and then was euthanized for owner’s willing” 

Furthermore, specify, please, how the metastases have been diagnosed.

Rows 193-200: this period is confusional, not convincing. Before you bring the hypothesis of eosinophilic panosteitis, and then, based on this hypothesis, you explain why it would be possible, but you do not have proof of it. It is confusing.

Row 196: the comma goes under the word hypothesis, not after the word findings

Rows 196-198: add, please, the citation for this sentence

Row 216: “it was to be ideal” is wrong, please correct

Rows 216-227: I understand what you want to explain, but the structure of this period is a bit incoherent. You talk, indeed, before about the dog, then you speak in general, then you return to the dog situation: this might be confusing for the reader. 

Row 233-235: the sentence “Unfortunately, owners refused standard adjuvant chemotherapy after the amputation and started metronomic chemotherapy” is not much linked to the previous one. Probably you need to delete “unfortunately” (otherwise you have to explain why the owner choice was not a good one) and just explain that as adjuvant chemotherapy the metronomic one, rather than the standard, has been preferred by the owner.

Rows 236-237: it is not clear to me what this sentence would say: “and it is difficult to predict if similar tumors may develop later in other sites affected by osteomyelitis” since your hypothesis is that the OSA developed in a site of previous osteomyelitis. Probably you meant something like “the development of new similar tumors can not be excluded later in other sites affected by osteomyelitis”.

Rows 227-228: add, please, the citation for this sentence

Row 239: I would write dogs instead of dog.

Rows 244-246:  This sentence “The incidence of relapse following an apparently “successful” treatment making its management challenging because the “cure” of the disease cannot be safely declared” is not clear. The incidence of relapse cannot be the subject of the sentence,  the verb tense seems wrong, and you shouldn’t need to use all these brackets. Can you re-formulate your thoughts, please?

Furthermore, in the discussion, can you formulate a hypothesis on how and when the dog may have come into contact and may have contracted the Serratia?

Figures:

Figure 2: the arrow, to me, is pointing at the trachea, not exactly the three ribs. I would add some asterisks on the ribs instead.

Figure 3: I suggest writing 2 distinct descriptions, one for figure 3a and one for figure 3b. Please, move the arrow toward the lesion you want to indicate. The arrows must touch exactly the “transition zone”.

Figure 4: move the arrows closer to what they are pointing at.

Author Response

REV 2: The discussion is a little confusing, not well structured with a mix of short sentences about different arguments. I suggest taking more care of the discussion. I also suggest an English review

Thank you very much, we have reformulated the discussions and carried out a revision of the English language for native English speaker.

Rows 12-13: “The tumor developed in the right humeral diaphysis one of the sites of polyostotic osteomyelitis.“ add a comma after the word diaphysis

Thank you. We added comma as suggested.

Rows 18-19: “Septic polyostotic osteomyelitis is a rare bone infection in dog, commonly caused by a long- term or persistent infection with fungal and bacterial agents”: add the meaning of ‘polyostotic’ inside the sentence (as Septic polyostotic osteomyelitis is a rare bone infection in dog, involving multiple bones and commonly caused by a long- term or persistent infection with fungal and bacterial agents)

Thank you. We added the meaning of “polyostotic” as suggested.

Rows 19-20: “a wide range of microorganisms could cause osteomyelitis, with bacterial infections being the most frequently reported”: add please a reference.

Thank you. We added the reference: Gieling, F.; Peters, S.; Erichsen, C.; Richards, R.G.; Zeiter, S.; Moriarty, TF. Bacterial osteomyelitis in veterinary orthopedics: Pathophysiology, clinical presentation and advances in treatment across multiple species. Vet J. 2019, 250, 44-54.

Row 23: delete “causative agent of ”

Thank you. We modified as suggested.

Row 26: I would write ‘with the disease’ instead of “ with osteomyelitis” to avoid the repetition.

Thank you. We modified as suggested.

Row 29: you should add some information about Serratia spp. as family, type of GRAM reactivity, and the natural environment for this bacteria (which represents an important nosocomial infection in humans)

Thank you. We added more details about Serratia spp. as suggested.

Row 34: by “osteomyelitis” do you mean “infectious osteomyelitis”?

Thank you. We added more details to the sentence as suggested.

Row 48, 57, 92: the name of the hospital is missing

Thank you. We added the hospital name as suggested.

Row 52 and following: can you add, please, the values of the normal range when you give the results of the chemistry?

Thank you. We added the values of normal range of the results of the chemistry cited

Row 56: write “Erlichia sp..”

Thank you. We modified as suggested.

Rows 58-59: I would write as follows “On clinical examination, the dog had no hyperthermia (38.7°C), exhibited signs of pain on deep palpation, together with soft tissue swelling, of left humerus and bilateral hypomyotrophy of both forelimbs.”

Thank you. We modified as suggested.

Row 64: I would write “the 5th, 6th and 7th right and left ribs” 

Thank you. We modified as suggested.

Row 64 (and legend of Figure 1, 3, and row 96): regarding the “sclerosis”, how can you detect and confirm that there is a deposit of fibrous tissue via X-ray? I’m not a radiologist, this is why I’m asking.

In radiology bone sclerosis means an abnormal increase in density of bone. Sclerosis is the localized formation of new bone and results in increased bone mass and occurs in response to several stimuli such as stress (e.g., subchondral sclerosis in osteoarthritis ), protection of a weakened area (e.g., sclerosis surrounding an osseous cystlike lesion or a subchondral bone cyst), walling off infection (e.g., adjacent to a sequestrum or in the medullary cavity adjacent to an area of osteomyelitis) These lesions are relatively common to be found on plain radiographs or CT scans.

Row 64: Please, add a comma after the word ‘reaction’.

Thank you. We added comma as suggested.

Rows 75, 76: spell out PO, BID and SID the first time you mention them in the text.

Thank you. Usually in clinical language these are internationally recognized acronyms. If you prefer, we could add their Latin meaning.

Rows 83-84: what do you mean with “abnormal neutrophils.”?

Thank you. We modified with high number of neutrophilis

Rows 90-91: it is not clear, to me, on what biological sample the PCR was performed to prove the negativity to Ehrlichia sp.

In the text: Bone marrow aspirates samples tested positive for Ehrlichia sp. using polymerase chain reaction (PCR).

Rows 94-95: do you mean that the left limb, not the right, appeared normal? 

Yes, the left limb appeared normal. We re-formulated the sentence correctly.

Row 98: the radiograph alone cannot establish that there is for sure a “large neoplastic osteogenesis”. The neoplastic nature of the lesion can be revealed and confirmed just at histology, the radiograph can eventually detect an area suggestive of neoplasia (as large presumptive neoplastic osteogenesis). Please, adjust the sentence.

Thank you. We modified as suggested.

Row 100: spell out CT the first time you mention it.

Thank you. We added this sentence as suggested.

Row 111: spell out NSAIDs

Thank you. We added this sentence as suggested. Usually in clinical language these are internationally recognized acronyms.

Rows 111-113: after how much time there was “The failure of medical therapy with NSAIDs, gabapentin and pamidronate disodium, and the progression of the lameness led to the decision to amputate the right forelimb”?

Thank you. We added “After 6-weeks of therapy”, the progression of the lameness led to the decision to amputate the right forelimb with owner’s willing.

Row 113: I would say “By histological examination of the entire bone”

Thank you. We modified this sentence as suggested.

Row 113-114: can you provide a brief description of the microscopic pattern and maybe add a picture of the histology? Eventually, the description can come only with the figure legend.

Thank you. unfortunately, the time elapsed from diagnosis exceeds the policy provided by the laboratory for archiving the histological blocks, so it is no longer possible to have histological photos. We have added the description of the microscopic pattern (Histological sample showed lamellar bone tissue with irregular islands of osteoid surrounded by malignant spindle to polygonal cells, organized in carpet or short bundles supported by scarce vascular stroma with productive extracellular osteoid matrix. Cartilage differentiation was appreciated multifocally in association with large areas of necrosis)

Row 116: write refused instead of refuse

Thank you. We modified this sentence as suggested.

Rows 115-117: reformulate, please, the sentence: “The dog undergoing a metronomic chemotherapy with cyclophosphamide (15 mg/m2/SID) and firocoxib (5 mg/Kg/SID) plus thalidomide at 10 mg/Kg, PO SID was”

Thank you. We modified this sentence as suggested.

Rows 117-118: I would write “8 months later the dog developed vertebral and thoracic metastases and then was euthanized for owner’s willing” 

Thank you. We modified this sentence as suggested.

Furthermore, specify, please, how the metastases have been diagnosed.

Thank you. We added this sentence as suggested.

Rows 193-200: this period is confusional, not convincing. Before you bring the hypothesis of eosinophilic panosteitis, and then, based on this hypothesis, you explain why it would be possible, but you do not have proof of it. It is confusing.

Thank you. We have deleted this hypothesis because we cannot scientifically prove the evidence of this

Row 196: the comma goes under the word hypothesis, not after the word findings

Thank you. We modified this sentence as suggested.

Rows 196-198: add, please, the citation for this sentence

Thank you. We deleted the period.

Row 216: “it was to be ideal” is wrong, please correct

Thank you. We modified this sentence as suggested.

Rows 216-227: I understand what you want to explain, but the structure of this period is a bit incoherent. You talk, indeed, before about the dog, then you speak in general, then you return to the dog situation: this might be confusing for the reader. 

Thank you. We modified this period

Row 233-235: the sentence “Unfortunately, owners refused standard adjuvant chemotherapy after the amputation and started metronomic chemotherapy” is not much linked to the previous one. Probably you need to delete “unfortunately” (otherwise you have to explain why the owner choice was not a good one) and just explain that as adjuvant chemotherapy the metronomic one, rather than the standard, has been preferred by the owner.

Thank you. We modified this sentence as suggested.

Rows 236-237: it is not clear to me what this sentence would say: “and it is difficult to predict if similar tumors may develop later in other sites affected by osteomyelitis” since your hypothesis is that the OSA developed in a site of previous osteomyelitis. Probably you meant something like “the development of new similar tumors can not be excluded later in other sites affected by osteomyelitis”.

Thank you. We modified this sentence as suggested.

Rows 227-228: add, please, the citation for this sentence

Thank you. We added this reference: Eiró, N.; Vizoso, F.J. Inflammation and cancer. World J. Gastrointest. Surg. 2012, 4(3), 62-72.

Row 239: I would write dogs instead of dog.

Thank you. We modified this sentence as suggested.

Rows 244-246:  This sentence “The incidence of relapse following an apparently “successful” treatment making its management challenging because the “cure” of the disease cannot be safely declared” is not clear. The incidence of relapse cannot be the subject of the sentence, the verb tense seems wrong, and you shouldn’t need to use all these brackets. Can you re-formulate your thoughts, please?

Thank you. We modified this sentence with: “The incidence of relapse following an apparently successful treatment remains high. The management of chronic osteomyelitis is challenging and a multidisciplinary approach involving radiologist, microbiologist and surgeon is required. Even following long periods of antibiotic treatment and surgical debridement, exacerbations can occur for many years and the cure of the disease cannot be safely declared.”

Furthermore, in the discussion, can you formulate a hypothesis on how and when the dog may have come into contact and may have contracted the Serratia?

We're sorry but we don't know how and when the dog has been infected by Serratia

Figures:

Figure 2: the arrow, to me, is pointing at the trachea, not exactly the three ribs. I would add some asterisks on the ribs instead.

Thank you. We added asterisks

Figure 3: I suggest writing 2 distinct descriptions, one for figure 3a and one for figure 3b. Please, move the arrow toward the lesion you want to indicate. The arrows must touch exactly the “transition zone”.

Thank you. We modified and writing two distinct descriptions.

Figure 4: move the arrows closer to what they are pointing at.

Thank you. We moved the arrows close to bone lesions. 

Reviewer 3 Report

In the manuscript entitled “Chondroblastic osteosarcoma associated with previous chronic osteomyelitis caused by Serratia liquefaciens in a German shepherd” the authors describe a case report in a German Shepherd dog. Below are the suggestions to improve the manuscript.

  1. Line 3-4: Chondroblastic osteosarcoma…….in a German Shepherd dog.
  2. The authors report that both Ehrlichia canis and Serratia liquefaciens were detected to be the causative agents of polyostotic osteomyelitis in a German Shepherd dog. However, they only attribute liquefaciens to be the main agent that caused chronic osteomyelitis. How did they arrive at this diagnosis? The authors should explain in detail.
  3. Line 209: Serratia marcescens is frequently isolated in human infections [24].
  4. What is the expanded form of OSA? The authors should expand all the abbreviations the first time.
  5. Did the authors perform histopathology? They should be displayed in the manuscript.

Author Response

Thank you very much, we have reported the changes, including them in the text.

  1. Line 3-4: Chondroblastic osteosarcoma…….in a German Shepherd dog.

Thank you. We modified the title of the case report modified as suggested.

  1. The authors report that both Ehrlichia canis and Serratia liquefaciens were detected to be the causative agents of polyostotic osteomyelitis in a German Shepherd dog. However, they only attribute liquefaciens to be the main agent that caused chronic osteomyelitis. How did they arrive at this diagnosis? The authors should explain in detail.

Thank you for your suggestion. We do not have stated that Erlichia and Serratia were detected to be the causative agents of polyostotic osteomyelitis in a German Shepherd dog. Ehrlichia canis does not cause osteomyelitis and bone biopsy culture revealed Serratia liquefaciens as the causative agent. Bone marrow aspirate samples were used only for the diagnosis of Ehrlichia sp.

  1. Line 209: Serratia marcescens is frequently isolated in human infections [24].

 Thank you. We modified the reference.

  1. What is the expanded form of OSA? The authors should expand all the abbreviations the first time.

Thank you. We added the expanded form of OSA, meaning osteosarcoma that is found in line 49 for the first time.

  1. Did the authors perform histopathology? They should be displayed in the manuscript.

Thank you. Unfortunately, the time elapsed from diagnosis exceeds the policy provided by the laboratory for archiving the histological blocks, so it is no longer possible to have histological photos. We have added the description of the microscopic pattern, as suggested by second reviewer

(Histological sample showed lamellar bone tissue with irregular islands of osteoid surrounded by malignant spindle to polygonal cells, organized in carpet or short bundles supported by scarce vascular stroma with productive extracellular osteoid matrix. Cartilage differentiation was appreciated multifocally in association with large areas of necrosis)
